# Removal of ECG Artifacts Affects Respiratory Muscle Fatigue Detection—A Simulation Study

**DOI:** 10.3390/s21165663

**Published:** 2021-08-23

**Authors:** Lorenz Kahl, Ulrich G. Hofmann

**Affiliations:** 1Drägerwerk AG & Co. KGaA, 23558 Lübeck, Germany; lorenz.kahl@draeger.com; 2Section for Neuroelectronic Systems, Department of Neurosurgery, Medical Center University of Freiburg, 79108 Freiburg, Germany; 3Faculty of Medicine, University of Freiburg, 79108 Freiburg, Germany

**Keywords:** biomedical signal processing, ECG, sEMG, EMG, neuromuscular fatigue, muscle fatigue, MNF, fApEn, SMR, fatigue detection, respiratory EMG, respiration, cardiogenic artifacts

## Abstract

This work investigates elimination methods for cardiogenic artifacts in respiratory surface electromyographic (sEMG) signals and compares their performance with respect to subsequent fatigue detection with different fatigue algorithms. The analysis is based on artificially constructed test signals featuring a clearly defined expected fatigue level. Test signals are additively constructed with different proportions from sEMG and electrocardiographic (ECG) signals. Cardiogenic artifacts are eliminated by high-pass filtering (HP), template subtraction (TS), a newly introduced two-step approach (TSWD) consisting of template subtraction and a wavelet-based damping step and a pure wavelet-based damping (DSO). Each method is additionally combined with the exclusion of QRS segments (gating). Fatigue is subsequently quantified with mean frequency (MNF), spectral moments ratio of order five (SMR5) and fuzzy approximate entropy (fApEn). Different combinations of artifact elimination methods and fatigue detection algorithms are tested with respect to their ability to deliver invariant results despite increasing ECG contamination. Both DSO and TSWD artifact elimination methods displayed promising results regarding the intermediate, “cleaned” EMG signal. However, only the TSWD method enabled superior results in the subsequent fatigue detection across different levels of artifact contamination and evaluation criteria. SMR5 could be determined as the best fatigue detection algorithm. This study proposes a signal processing chain to determine neuromuscular fatigue despite the presence of cardiogenic artifacts. The results furthermore underline the importance of selecting a combination of algorithms that play well together to remove cardiogenic artifacts and to detect fatigue. This investigation provides guidance for clinical studies to select optimal signal processing to detect fatigue from respiratory sEMG signals.

## 1. Introduction

The over-exaggerated use of skeletal muscles leads to fatigue recognizable in the corresponding surface electromyographic (sEMG) signals [1,2,3]. Respiratory muscles are no exceptions from fatiguing [4,5,6], except it is rarely seen in everyday life.

Unfortunately, in the case of mechanical ventilation of the critically ill patient, fatigued respiratory muscles are related to a pathway leading to respiratory failure [7]. Respiratory failure is a clinically relevant issue in connection with weaning a patient from mechanical ventilation, especially if this process fails. Weaning failure is associated with an imbalance of neuromuscular capacity and ventilatory needs [8]. Although this failure is in many cases of a multi-factorial nature [8], it is often accompanied with respiratory muscle fatigue. Brochard et al. [9] demonstrated that unsuccessful weaning trials in patients meeting the usual weaning criteria were associated with diaphragmatic fatigue. It is thus desirable to monitor and predict respiratory muscle fatigue based on sEMG, especially when weaning is the next therapeutic step.

Until now, automated analysis of respiratory sEMG from the diaphragm and intercostal muscles has been hampered by strong contamination with cardiogenic artifacts. In the context of this study, cardiogenic artifacts refer to disturbances in the respiratory sEMG signal originating from the electrical activity of the heart. Prior to any further evaluation, these artifacts have to be compensated for, while fatigue-related features in the sEMG have to be conserved.

An early approach to determine muscular fatigue based on respiratory EMG by Sieck et al. [5] included the detection of QRS segments with a Schmitt trigger and their omission from the corresponding EMG segments. QRS segments were also excluded by Brochard et al. [9]. To this end, Sinderby et al. [10] proposed analyzing only selected signal segments between two subsequent R peaks. They suggested utilizing only signal segments from between 50% and 75% of the RR interval for fatigue analysis and detrended selected segments. Ortega et al. [11] proposed utilizing a fifth order recursive least squares (RLS) filter to produce an artificial electrocardiographic (ECG) signal. This artificial signal claims to cancel cardiogenic interferences by subtraction. Template subtraction [12] is a common approach to eliminate cardiogenic artifacts for general respiratory sEMG processing. It utilizes the repetitive characteristics of cardiogenic artifacts for their removal and is also known as event-synchronous cancellation [13,14]. A template is gained by averaging the contaminated signal in segments corresponding to each heartbeat aligned at QRS times. This template is used in an adaptive noise cancellation scheme by subtracting it from the contaminated signal repetitively aligned with heartbeats. A study comparing different elimination methods for cardiogenic artifacts in respiratory sEMG was recently published by Petersen et al. [15].

The study at hand compares different methods to eliminate cardiogenic artifacts from sEMG signals and evaluates the impact of residual cardiogenic artifacts on a subsequently performed fatigue detection. We introduce a novel two-step approach consisting of additive template subtraction and a subsequent wavelet-based, multiplicative damping step. To scrutinize our methods, signals with specific properties regarding strength and characteristics of cardiogenic artifacts are synthesized. The synthetic test signals are not intended to be a completely realistic model for respiratory sEMG. Instead, they were chosen to include two specific characteristics: (1) cardiogenic components with changing shapes and (2) fatigue-related features corresponding to a known fatigue state. They are utilized to evaluate a set of fatigue detection algorithms for their resistance against residues from cardiogenic artifact removal. We specifically seek to optimize algorithms in order to minimize the impact of cardiogenic artifacts on the fatigue analysis.

## 2. Methods

The following section describes the synthesis of artificial sEMG signals containing cardiogenic artifacts. Afterwards, we describe methods to eliminate these artifacts and algorithms to determine muscular fatigue. Finally, we introduce criteria to evaluate the fatigue detection performance.

### 2.1. Construction of Artificial Signals

Surface electromyographic signals (sEMG) were recorded in two previous studies [16,17] during contractions of the upper arm with different load levels. Load levels are stated as fractions of an initially performed maximum voluntary contraction (MVC). Subjects were instructed to perform contractions with 20%, 40% and 60% of their individual MVC for 3 min. They were allowed to cancel earlier if feeling fatigued. In between tasks, there was a resting period of at least 30 min. For this investigation, nine subjects displaying a clear difference in fatigue indexes at 20% and 60% MVC level provided sEMG recordings. For this study, only the first minute of sEMG signals was used. Raw sEMG signals were high-pass filtered with a third order Butterworth filter with 1 Hz corner frequency to remove baseline offsets. The resulting signals with zero mean are denoted as ${\mathrm{EMG}}_{20}\left(t\right)$ and ${\mathrm{EMG}}_{60}\left(t\right)$.
An exemplary signal segment of ${\mathrm{EMG}}_{60}\left(t\right)$ is displayed in Figure 1 (subpanel A).

To mimic respiration-like activity, sEMG signals were modulated by a respiration pattern consisting of 1 s inspiration with full muscular activity followed by 3 s of expiration with 30% muscular activity. This corresponds to a respiratory rate of 15 breaths per minute that can be considered as typical for an adult at rest [18]. Furthermore, both inspiration and expiration duration are within the normal range for a resting subject [19]. The residual activity of 30% in expirations was chosen to ensure that the fatigue algorithms are continuously supplied with sEMG data featuring realistic fatigue related properties. The activity changes smoothly between transitions. See Figure 1 (subpanel B) for a visualization of the activity pattern ${\mathrm{MS}}_{M}\left(t\right)$. The modulated sEMG signal is obtained by:
(1)$${\mathrm{EMG}}_{\Lambda }^{M}\left(t\right)\;=\;{\mathrm{MS}}_{M}\left(t\right)\;\cdot \;{\mathrm{EMG}}_{\Lambda }\left(t\right)$$
where Λ denotes the MVC load level.
An example is shown in Figure 1 (subpanel C).

ECG signals were sourced from the CARDIODAT dataset [20] (accessed via Physionet [21]). Channels I and II were slightly upsampled from 1 k
Hz to 1024 Hz, matching the sample rate of the sEMG signals. Linear interpolation was utilized for the upsampling. They are denoted as ${\mathrm{ECG}}_{I}\left(t\right)$ and ${\mathrm{ECG}}_{\mathrm{II}}\left(t\right)$. A baseline filtering removes the arbitrary offset from the ECG signals and yields ${\mathrm{ECG}}_{I}^{\mathrm{BLF}}\left(t\right)$ and ${\mathrm{ECG}}_{\mathrm{II}}^{\mathrm{BLF}}\left(t\right)$.

Often, ECG signals already include electromyographic components from respiratory muscles [22]. We found that this is also true in our case. However, these components do not feature the intended fatigue condition and would impair the subsequent evaluation. Among others, a Savitzky–Golay filter was suggested for denoising ECG signals [22,23]. Thus, a Savitzky–Golay filter [24] was applied to ${\mathrm{ECG}}_{I}^{\mathrm{BLF}}\left(t\right)$ and ${\mathrm{ECG}}_{\mathrm{II}}^{\mathrm{BLF}}\left(t\right)$ to attenuate these EMG-like signal components. To account for high frequencies within the QRS complex, different parameters of the filter were utilized. For signal parts within or close to the QRS segment (from 0.05 s prior to the time of QRS detection until 0.1 s after), the order was 5 and the length was 15 bins. Otherwise, a Savitzky–Golay filter of order 3 and length 25 was applied. The Savitzky–Golay filtered signals are denoted as ${\mathrm{ECG}}_{I}^{\mathrm{SG}}\left(t\right)$ and ${\mathrm{ECG}}_{\mathrm{II}}^{\mathrm{SG}}\left(t\right)$, which are exemplified in Figure 1D,E. The Savitzky–Golay filter’s effect is visualized in the Appendix B Figure A1.

Heartbeat artifacts contaminating respiratory sEMG usually show a much larger irregularity compared to the ECG features from channel I and II. This effect is caused by variations in electrode positions. The position of channels I and II are much less prone to changes in geometry between the heart and electrode due to breathing activities. Electrode positions close to the diaphragm on the costal margin are much more influenced by varying separations between the heart and electrodes. The same effect can also be observed in respiratory EMG recordings with esophageal electrodes [10]. In order to mimic a similar effect in our synthetic signals, we decided to mix ECG channels I and II with varying weights resembling the breathing phase. These peripheral leads seemed advantageous to us because they are bipolar leads and are not susceptible to intrinsically include much respiratory EMG and/or ECG irregularity caused by these varying separations between heart and electrodes. For respiratory sEMG, it is common practice to use a bipolar recording with electrodes bilaterally distributed on the rib cage [25]. Although precordial ECG leads such as V4 also feature a location that is prone to similar effects, we have refrained from using it. The main reason is that V4 is an unipolar lead [26]. Additionally, the artificially mixed channel from lead I and II enabled us to control the timing to be consistent with the breathing phase pattern. Intrinsically included electromyographic components and/or ECG irregularity from the ECG signal would not match the temporal pattern of our artificial breathing phase pattern. The weighting signal ${\mathrm{WS}}_{C}\left(t\right)$ was constructed to reflect the changing lung volume. It features a steep ascent at the beginning of the inspiration and a slower decay-like descent in the expiration. The artificially mixed ECG channel (denoted channel A) was calculated as: (2)$${\mathrm{ECG}}_{A}\left(t\right)=\left(1-{\mathrm{WS}}_{C}\left(t\right)\right)\;\cdot \;{\mathrm{ECG}}_{I}^{\mathrm{SG}}\left(t\right)+{\mathrm{WS}}_{C}\left(t\right)\;\cdot \;{\mathrm{ECG}}_{\mathrm{II}}^{\mathrm{SG}}\left(t\right)$$

An example of this mixing is shown in Figure 1 (subpanel G). An evaluation of the increased heartbeat irregularity is appended in the Appendix B Figure A2.

Artificial signals were finally constructed by additively combining preprocessed ${\mathrm{ECG}}_{A}\left(t\right)$ with modulated ${\mathrm{EMG}}_{20}^{M}\left(t\right)$ and ${\mathrm{EMG}}_{60}^{M}\left(t\right)$ signals. The combination was performed with different artifact magnitude values in relation to the sEMG power. To quantify this relation, we used the EMG level η known from [27]. It reflects the ratio of RMS sEMG power (of the unmodulated sEMG signal) and peak-to-peak QRS amplitude. The peak-to-peak QRS amplitude was always calculated purely based on ECG channel I even if an ECG from a mixture of channels is used. This assured a constant absolute sEMG power across different ECG channels.

All this yields to the following equation for the synthesis of artificial test signals based on the MVC load level Λ: (3)$${\mathrm{ATS}}_{\Lambda ,\eta ,A}\left(t\right)\;=\;{\mathrm{ECG}}_{A}\left(t\right)+\eta \;\cdot \;\frac{{\mathrm{QRS}}_{\mathrm{minmax}}\left({\mathrm{ECG}}_{I}^{\mathrm{SG}}\left(t\right)\right)}{\mathrm{RMS}\left({\mathrm{EMG}}_{\Lambda }\left(t\right)\right)}\;\cdot \;{\mathrm{EMG}}_{\Lambda }^{M}\left(t\right)$$

Artificial test signals were generated with η = 0.01,0.02,0.05,0.1and0.2 and MVC load levels Λ of 20% and 60%. An example of ${\mathrm{ATS}}_{60,0.05,A}\left(t\right)$ can be found in Figure 1 (subpanel H).

Simplified test signals ${\mathrm{ATS}}_{\Lambda ,\eta ,I}\left(t\right)$ that are purely based on the ECG channel I were constructed for comparison purposes.
(4)$${\mathrm{ATS}}_{\Lambda ,\eta ,I}\left(t\right)={\mathrm{ECG}}_{I}^{\mathrm{SG}}\left(t\right)+\eta \;\cdot \;\frac{{\mathrm{QRS}}_{\mathrm{minmax}}\left({\mathrm{ECG}}_{I}^{\mathrm{SG}}\left(t\right)\right)}{\mathrm{RMS}\left({\mathrm{EMG}}_{\Lambda }\left(t\right)\right)}\;\cdot \;{\mathrm{EMG}}_{\Lambda }^{M}\left(t\right)$$

Pure sEMG signals ${\mathrm{RSM}}_{\Lambda }\left(t\right)$ and ${\mathrm{RS}}_{\Lambda }\left(t\right)$ were also constructed for comparison purposes. They featured the same sEMG power as for η=0.2 but included no ECG contamination. Because all utilized fatigue detection algorithms are invariant to sEMG scaling, it was sufficient to construct only comparison signals with a single sEMG power.

The first signal ${\mathrm{RSM}}_{\Lambda }\left(t\right)$ corresponds to the reference case and also incorporates the sEMG modulation.
(5)$${\mathrm{RSM}}_{\Lambda }\left(t\right)=0.2\;\cdot \;\frac{{\mathrm{QRS}}_{\mathrm{minmax}}\left({\mathrm{ECG}}_{I}^{\mathrm{SG}}\left(t\right)\right)}{\mathrm{RMS}\left({\mathrm{EMG}}_{\Lambda }\left(t\right)\right)}\;\cdot \;{\mathrm{EMG}}_{\Lambda }^{M}\left(t\right)$$

As no ECG contamination is included, the reference case also skips any cardiogenic artifact removal. However, it contains the same sEMG components. The reference case is used for the normalization of fatigue detection results and at the subsequent evaluation step. It is used to determine how much the fatigue detection results are impaired by ECG contamination and elimination methods of these cardiogenic artifacts. ${\mathrm{RSM}}_{\Lambda }\left(t\right)$ is used to calculate the reference fatigue indexes $\Xi_{\Lambda }\left(t\right)$ (see below).

The second signal for comparison purposes is ${\mathrm{RS}}_{\Lambda }\left(t\right)$. It is a scaled version of the pure static sEMG without modulation and is used to calculate the fatigue indexes $\Psi_{\Lambda }\left(t\right)$ (see below).
(6)$${\mathrm{RS}}_{\Lambda }\left(t\right)=0.2\;\cdot \;\frac{{\mathrm{QRS}}_{\mathrm{minmax}}\left({\mathrm{ECG}}_{I}^{\mathrm{SG}}\left(t\right)\right)}{\mathrm{RMS}\left({\mathrm{EMG}}_{\Lambda }\left(t\right)\right)}\;\cdot \;{\mathrm{EMG}}_{\Lambda }\left(t\right)$$

See Figure 2 column I for a visualization of the reference signal ${\mathrm{RSM}}_{\Lambda }\left(t\right)$ and artificial test signals ${\mathrm{ATS}}_{\Lambda ,\eta ,A}\left(t\right)$. A block diagram visualizing the test signal generation and a table stating which sEMG and ECG records being combined can be found in the Appendix B Figure A3 and Figure A4. A data file with a corresponding R-script can be found in the Appendix A to obtain various signals ${\mathrm{ATS}}_{\Lambda ,\eta ,\mathrm{ch}}\left(t\right)$, ${\mathrm{RSM}}_{\Lambda }\left(t\right)$ and ${\mathrm{RS}}_{\Lambda }\left(t\right)$.

### 2.2. Removal of Cardiogenic Artifacts

Template subtraction [28,29] was used to reduce cardiogenic artifacts. This method utilizes the repetitive nature of ECG and is also known as an event synchronous noise canceller [14]. Past signal segments covering one heartbeat are aligned along the QRS complex. The time alignment of QRS complexes is based on the QRS detector and time refinement described in the following two paragraphs. A template heart segment was constructed from the average of superimposed segments. The procedure was performed with signals upsampled to 2048 Hz. This allowed a more precise time location of the template, as the template is placed relative to sample time points and cannot be placed in between. The resulting signals were downsampled back to 1024 Hz.

Structural intensity (SI), as proposed in [30], was used to detect QRS events from the artificial signals. The algorithm is based on the second derivative of the signal. Extrema are localized in low pass filtered versions corresponding to different scales. Corresponding extrema over different scales are connected by extrema lines. A function $G_{m}^{***}\left(t\right)$ evaluates these extrema lines and yields a scalar signal over time. Peaks of $G_{m}^{***}\left(t\right)$ above a threshold are considered to correspond to a QRS complex. The threshold is chosen at half of the maximal value of $G_{m}^{***}\left(t\right)$. Instead of an SI scheme with 14,000 scales (as suggested by [30]), we used an adopted scheme with a reduced set of 28 scales, as already reported in a previous publication [27]. Furthermore, we also made use of the time alignment fine tuning from [27]. This intermediate exact time alignment step refines the time location of the structural intensity-based QRS detector. A better time alignment enables the construction of a more precise template. The template can thus be subtracted more precisely, reducing the residual cardiogenic artifact.

A raw template is constructed from the superimposed heart cycle segments by calculating the mean. The raw template is filtered with a Savitzky–Golay filter of order 6 and length 25. Optionally, the template is treated with an additional denoising step. The template is wavelet transformed by a stationary wavelet transform (SWT) with 6 scales and the Daubechies wavelet with 5 vanishing moments [31] as the mother wavelet. The first three levels that hold the highest frequency components are set to zero. Levels four to six are set to zero in ranges that feature an RMS power below the median RMS power of that scale. Ranges in levels are reduced if the RMS power is in between the median RMS and twice the median RMS to assure smooth transitions. In this way, it is assured that higher frequencies are kept within the QRS segment but voided otherwise. If a range is set to zero, it is furthermore assured that the same range is also set to zero in all earlier levels. The denoised template is gained by the inverse SWT of the modified levels. The final template is trimmed to start 0.3 s prior to the time of QRS detection. It has the length of the average heart cycle. The segment is tapered by multiplication with a trapezoid window. The rising flank has a length of 0.1 s, while the declining flank is 0.2 s long. The template is subtracted iteratively, starting at the beginning of the signal. The signal yielded from the subtraction is post-processed with a third order Butterworth high-pass filter with corner frequency of 15 Hz or alternatively 35 Hz. Signals with reduced cardiac artifacts are denoted ${\mathrm{ATS}}_{\Lambda ,\eta ,\mathrm{ch}}^{\mathrm{cr}}\left(t\right)$; “cr” identifies the utilized artifact removal method. TS15 and TS35 denote the ordinary template subtraction with an additional high-pass filter with 15 Hz and 35 Hz corner frequency, respectively. TSW15 and TSW35 identify the application of the wavelet denoised template. See Figure 2 column II for exemplary signals. The sole application of the additional high-pass filter to suppress cardiogenic artifacts is listed as HP15 and HP35, respectively, for comparison purposes.

Due to irregularities in the QRS complex, the ordinary template subtraction variants are not able to completely eliminate the cardiogenic artifacts. Exemplary residual artifacts are shown in Figure 1 subpanel I). In our test signals, this effect is provoked by the increased heartbeat irregularity in the artificially mixed ECG channel. To reduce the remaining cardiogenic artifact, we introduced a wavelet-based damping step to be carried out subsequently to the template subtraction. In this second step, the signal ${\mathrm{ATS}}_{\Lambda ,\eta ,\mathrm{ch}}^{\mathrm{TSW}15}\left(t\right)$ resulting from the ordinary template subtraction is wavelet transformed by a stationary wavelet transform (SWT). We used 8 scales and the Daubechies wavelet again with 5 vanishing moments [31] as the mother wavelet. For each SWT level, *i* a signal ${\mathrm{ATS}}_{\Lambda ,\eta ,\mathrm{ch}}^{\mathrm{TSW}15\_\mathrm{SWT},i}\left(t\right)$ is obtained. An average remaining cardiogenic artifact signal ${\mathrm{wARCA}}_{\Lambda ,\eta ,\mathrm{ch}}^{\mathrm{TSW}15\_\mathrm{SWT},i}\left(\tau_{c}\right)$ is constructed by averaging all rectified heart cycle segments of ${\mathrm{ATS}}_{\Lambda ,\eta ,\mathrm{ch}}^{\mathrm{TSW}15\_\mathrm{SWT},i}\left(t\right)$ at the same relative heart cycle time $\tau_{c}$. A damping template $\mathrm{dTemp}\left(\tau_{c},i\right)=\frac{m_{\tau }}{{\mathrm{wARCA}}_{\Lambda ,\eta ,\mathrm{ch}}^{\mathrm{TSW}15\_\mathrm{SWT},i}\left(\tau_{c}\right)}$ is subsequently constructed with a median value $m_{\tau }=\underset{\tau_{c}}{\mathrm{median}}\left({\mathrm{wARCA}}_{\Lambda ,\eta ,\mathrm{ch}}^{\mathrm{TSW}15\_\mathrm{SWT},i}\left(\tau_{c}\right)\right)$. The damping template value $\mathrm{dTemp}\left(\tau_{c},i\right)$ is set to one if ${\mathrm{wARCA}}_{\Lambda ,\eta ,\mathrm{ch}}^{\mathrm{TSW}15\_\mathrm{SWT},i}\left(\tau_{c}\right)<\theta_{i}$, with the threshold $\theta_{i}\;=\;\left(1+\frac{8-i}{16}\right)\;\cdot \;m_{\tau }$. In the low wavelet levels (holding the relevant frequency ranges for the subsequent fatigue detection), the threshold $\theta_{i}$ is well above the values of the wavelet level outside of remaining cardiogenic artifacts (concentrated around $m_{\tau }$). The selection of the median instead of a mean value assures that $m_{\tau }$ will be quite invariant of the magnitude of the remaining cardiogenic artifact. This holds as long as the time duration of the remaining artifact is short compared to the total length of the cardiac cycle. Luckily, this is usually the case because the remaining cardiogenic artifacts are usually concentrated within the QRS range. In the unlikely case that there are no remaining cardiogenic artifacts, this independence of $\theta_{i}$ from the artifact magnitude assures that there is hardly any manipulation within the relevant levels.

The damping template $\mathrm{dTemp}\left(\tau_{c},i\right)$ is multiplied synchronously to the QRS time with ${\mathrm{ATS}}_{\Lambda ,\eta ,\mathrm{ch}}^{\mathrm{TSW}15\_\mathrm{SWT},i}\left(t\right)$ to obtain the modified wavelet level ${\mathrm{ATS}}_{\Lambda ,\eta ,\mathrm{ch}}^{\mathrm{TSW}15\_\mathrm{SWT}\mathrm{mod},i}\left(t\right)$. This is in contrast to the regular template subtraction, where the template is QRS synchronously subtracted. The final cardiogenic artifact eliminated signal ${\mathrm{ATS}}_{\Lambda ,\eta ,\mathrm{ch}}^{\mathrm{TSWD}15}\left(t\right)$ is obtained by the application of the inverse SWT of the modified levels ${\mathrm{ATS}}_{\Lambda ,\eta ,\mathrm{ch}}^{\mathrm{TSW}15\_\mathrm{SWT}\mathrm{mod},i}\left(t\right)$. TSWD15 and TSWD35 denote this adopted template subtraction with an additional damping step. In contrast to the description of TSWD15 above, TSWD35 utilizes an 35 Hz high-pass filter. A block diagram visualizing the removal of cardiogenic artifacts can also be found in the Appendix B Figure A5.

A two-step approach to further reduce cardiogenic artifacts that could not be eliminated by one method is also known from Abbaspour and Fallah [32]. Our approach differs in two key aspects besides the fact that they used an artificial neural network (ANN) to obtain an estimation of ECG for subtraction in the first step instead of ordinary template subtraction. The first aspect concerns the elimination of artifacts. Abbaspour and Fallah [32] set the wavelet coefficient to zero, if it exceeds a threshold. This corresponds to wavelet denoising with hard thresholding. In contrast, we chose a damping approach that also allowed for a wavelet coefficient to be reduced by a fraction. Thus, frequency components can be decreased by adjustable degrees instead of a binary decision to keep or disregard them. The second aspect concerns the consideration of the phase of the cardiac cycle. Instead of a fixed regime, we adopt our reduction scheme depending on the heart cycle time $\tau_{c}$. This is to make the algorithm remove artifacts more easily if they are very likely (e.g., within the QRS segment). On the other hand, the hurdles for the removal of an artifact are higher if it is located in segments where cardiogenic artifacts are very unlikely. In contrast, Abbaspour and Fallah [32] do not adopt the threshold depending on the heart phase.

For comparison purposes, a variant that only consists of the damping step is displayed as well. In this variant, the stationary wavelet transform is performed directly on the input signal ${\mathrm{ATS}}_{\Lambda ,\eta ,\mathrm{ch}}\left(t\right)$ without any manipulations. The template subtraction utilized in TSWD15 and TSWD35 is skipped. This cardiogenic artifact elimination method is denoted as DSO (damping step only). A visualization of an exemplary average remaining cardiogenic artifact signal ${\mathrm{wARCA}}_{\Lambda ,\eta ,\mathrm{ch}}^{\mathrm{TSW}15\_\mathrm{SWT},i}\left(\tau_{c}\right)$, a damping template $\mathrm{dTemp}\left(\tau_{c},i\right)$ and exemplary results for DSO, TSW and TSWD can be found in the Appendix B Figure A6.

Ordinary template subtraction cannot completely erase cardiogenic artifacts [33]. Deng et al. [33] state that theoretically required assumptions for template subtraction are not completely met in practical applications. It can be assumed that these issues—although to a lesser extent—also occur with other methods. The remaining artifacts, however, are mainly concentrated in the vicinity of former QRS complexes. One possible strategy to cope with those remaining artifacts is gating. Gating ensures that signal segments around the QRS are excluded from further evaluation. For each of the methods to remove cardiogenic artifacts (as described above), we added variants combining them with gating. We applied a fixed length gating starting 0.05 s before the detected QRS and extending 0.1 s beyond.

### 2.3. Fatigue Detection

Muscle fatigue detection was performed epoch-wise. Epoch lengths $L_{e}$ of 0.125, 0.25, 0.5 and 1 s corresponding to epoch sizes $N_{e}$ of 128, 256, 512 and 1024 bins were used. Regardless of the employed epoch size, a fatigue value was calculated every 0.125 s establishing fatigue index signals $\Phi \left(t\right)$ with 8 Hz. Epochs were overlapping in the case of $L_{e}\geq 0.25\;s$. In the case no QRS gating is applied (gating strategy **GN**), the epoch is constructed regardless of possible gating. A gating signal is completely disobeyed. All possible remaining QRS artifacts are included in the following fatigue detection. If gating is in place (gating strategy **GO**), the gated segments are omitted in the epoch. Even in this case, a fatigue index value is calculated every 0.125 s corresponding to 8 Hz. Next to the well-known mean frequency (MNF) [34], we decided to use a spectral moments ratio of order five (SMR5) [35] and fuzzy approximate entropy (fApEn) [36] as fatigue detection algorithms. Both latter algorithms scored very good in a general comparison of sEMG-based fatigue detection algorithms [17]. The estimation of the power spectral density (PSD) for both MNF and SMR5 was performed by nonparametric (Welch with different numbers of subsegments $k_{w}$) as well as parametric methods (Burg) [37]. To account for the remaining low-frequency cardiogenic artifacts, we tried different PSD lower bound $b_{l}$ to exclude low-frequency bins. The Appendix B contains further details about the utilized fatigue detection. A block diagram for the fatigue detection is depicted in Figure A7. Figure A8 includes algorithmic details of the fatigue detection algorithms and power spectral density estimation methods. An exemplary visualization of epoch construction and gating is shown in Figure A1.

Various fatigue index signals $\Phi_{\Lambda ,\eta ,\mathrm{ch}}^{\mathrm{cr},\mathrm{gs},N_{e},\mathrm{fd}}\left(t\right)$ were calculated for each artifact reduction method. Based on artifact reduced signals ${\mathrm{ATS}}_{\Lambda ,\eta ,\mathrm{ch}}^{\mathrm{cr}}\left(t\right)$, different combinations of gating strategies gs, epoch size $N_{e}$ and fatigue detection methods fd were generated. In addition, the reference fatigue index signals $\Xi_{\Lambda }^{\mathrm{gs},N_{e},\mathrm{fd}}\left(t\right)$ are obtained from the signals ${\mathrm{RSM}}_{\Lambda }\left(t\right)$. Fatigue index signals $\Psi_{\Lambda }^{\mathrm{gs},N_{e},\mathrm{fd}}\left(t\right)$ result from the unmodulated and uncontaminated pure sEMG signals ${\mathrm{RS}}_{\Lambda }\left(t\right)$. Both the reference fatigue signal $\Xi_{\Lambda }^{\mathrm{gs},N_{e},\mathrm{fd}}\left(t\right)$ and the signal $\Psi_{\Lambda }^{\mathrm{gs},N_{e},\mathrm{fd}}\left(t\right)$ were not based on signals treated by any cardiogenic artifact removal method. As no QRS is present, gating is not necessary and was generally not applied to the reference signals (this corresponds to gating strategy GN). If gating is applied for comparison analysis, the gating time information is taken from the pure ECG signal.

Fatigue index signals calculated with smaller epoch sizes have a generally higher disturbance (for definition see Section 2.4). A longer epoch size leads to smaller disturbances due to a higher averaging effect. To enable an unbiased comparison, all epoch sizes are combined by a moving average to a smoothed signal. The length of the moving average filter is chosen in a way that each fatigue sample will incorporate information from a 1 s segment of the sEMG signal. These filtered fatigue index signals $\Phi_{\Lambda ,\eta ,\mathrm{ch}}^{\mathrm{cr},\mathrm{gs},N_{e},\mathrm{fd},\mathrm{MAfilt}}\left(t\right)$ were used in the following evaluation only when comparing different epoch sizes.

Fatigue indexes based on different fatigue algorithms come in different ranges. In order to compare the actual performance, the fatigue index signals $\Phi \left(t\right)$ are normalized, as described in [17]. Normalization is based on a linear regression line $G_{60}^{\mathrm{gs},N_{e},\mathrm{fd}}\left(t\right)$ that is obtained from the segment 0 s<t<60 s of the reference fatigue index signal $\Xi_{60}^{\mathrm{GN},N_{e},\mathrm{fd}}\left(t\right)$ originating from the 60% MVC load case. No gating (strategy GN) was utilized for the reference signals regardless of the gating strategy for the fatigue index signal $\Phi \left(t\right)$ undergoing normalization. All other relevant parameters, such as epoch size $N_{e}$ and applied fatigue index fd, are chosen to match. The offset at t=0 s and the change in the first minute of the linear regression line $G_{60}^{\mathrm{gs},N_{e},\mathrm{fd}}\left(t\right)$ were used to obtain normalized fatigue index signals $\hat{\Phi }\left(t\right)$, as given by Equation (Equation 7).
(7)$${\hat{\Phi }}_{\Lambda ,\eta ,\mathrm{ch}}^{\mathrm{cr},\mathrm{gs},N_{e},\mathrm{fd}}\left(t\right)=\frac{\Phi_{\Lambda ,\eta ,\mathrm{ch}}^{\mathrm{cr},\mathrm{gs},N_{e},\mathrm{fd}}\left(t\right)-G_{60}^{\mathrm{gs},N_{e},\mathrm{fd}}\left(0\;s\right)}{G_{60}^{\mathrm{gs},N_{e},\mathrm{fd}}\left(60\;s\right)-G_{60}^{\mathrm{gs},N_{e},\mathrm{fd}}\left(0\;s\right)}$$

The normalized fatigue index signals $\hat{\Xi }\left(t\right)$ and $\hat{\Psi }\left(t\right)$ are obtained in an analogous manner. Accordingly, normalization should arrange for the normalized fatigue index of the 60% MVC load case evolving from 0 to 1 within the considered minute regardless of their primary value range. As fatigue index signals from 20% MVC load case do not show fatigue, their normalized values are rather constant around 0. Exemplary fatigue index signals are shown in Figure 3 column III.

### 2.4. Evaluation of Quality

In a first intermediate step, different measures were calculated to evaluate the deviation of the ECG “cleaned” signals compared to the pure scaled EMG components in the time and frequency domain. These measures include error values based on the electromyographic signals themselves, their envelope signals and the corresponding power spectral densities. Since these measures represent only a secondary result, their description and the corresponding detailed results are outsourced to Figure A9 in the Appendix B.

The overall aim is to find a signal processing chain suitable for fatigue detection in sEMG signals contaminated with cardiogenic artifacts. The signal processing chain consists of the removal of cardiogenic artifacts and subsequent fatigue detection. For the evaluation of different signal processing chain variants, the main criterion is the invariance of the fatigue results at different strengths of ECG contamination compared to the fatigue results of the reference case without cardiogenic artifacts. The fatigue results should change as little as possible despite the presence of cardiogenic artifacts. Different strengths of ECG contamination are represented by signals with different EMG levels η. This invariance is the basis for the assessment of the quality of the cardiogenic artifact removal and the consecutive fatigue detection according to the three criteria presented below.

The first criterion (A) targets the desired invariance of the fatigue algorithm’s result despite the presence of cardiogenic artifacts. The impact of cardiogenic artifacts on the fatigue index ${\hat{\Phi }}_{60,\eta ,\mathrm{ch}}^{\mathrm{cr},\mathrm{gs},N_{e},\mathrm{fd}}$ is quantified by the RMS power of the difference to the reference fatigue index ${\hat{\Xi }}_{60}^{\mathrm{GN},N_{e},\mathrm{fd}}$. Only the fatigue index signals from the 60% MVC load cases are considered. Criterion A is defined as: (8)$$\gamma_{A}\left(\eta ,\mathrm{ch},\mathrm{cr},\mathrm{gs},N_{e},\mathrm{fd}\right)=\sqrt{\frac{1}{60}\int_{0}^{60}{\left({\hat{\Phi }}_{60,\eta ,\mathrm{ch}}^{\mathrm{cr},\mathrm{gs},N_{e},\mathrm{fd}}\left(t\right)-{\hat{\Xi }}_{60}^{\mathrm{GN},N_{e},\mathrm{fd}}\left(t\right)\right)}^{2}dt}$$

Another desirable characteristic of fatigue signal processing is the separability of fatiguing and non-fatigued situations. This separability should also be achieved in the presence of cardiogenic artifacts. The separability (denoted criterion B) is evaluated based on the last 15 s of the 20% and 60% MVC load cases. Vectors containing the fatigue index values of these last 15 s are defined as $\bar{\Phi }\left(\Lambda ,\eta ,\mathrm{ch},\mathrm{cr},\mathrm{gs},N_{e},\mathrm{fd}\right)={\hat{\Phi }}_{\Lambda ,\eta ,\mathrm{ch}}^{\mathrm{cr},\mathrm{gs},N_{e},\mathrm{fd}}\left(t\right){|}_{45\;s<t<60\;s}$. The values from $\bar{\Phi }$ based on the 60% MVC load case should clearly indicate fatigue with normalized values close to one. In contrast, no fatigue is expected at the 20% MVC load case, leading to normalized fatigue values around zero. The separability is based on the overlap between distributions of fatigue values from both load cases. See Figure 3 for an example. To avoid an estimation of the probability density functions (PDF) of $\bar{\Phi }$, the maximum distance of their cumulative density function (CDF) is calculated. This maximum distance is equal to the difference of one to the overlapping area of PDFs and also known as the test statistic from the Kolmugorov–Smirnov test [38]. The separability is defined as: (9)$$\gamma_{B}\left(\eta ,\mathrm{ch},\mathrm{cr},\mathrm{gs},N_{e},\mathrm{fd}\right)=\max_{\phi }\left|F_{\bar{\Phi }\left(20,\eta ,\mathrm{ch},\mathrm{cr},\mathrm{gs},N_{e},\mathrm{fd}\right)}\left(\phi \right)-F_{\bar{\Phi }\left(60,\eta ,\mathrm{ch},\mathrm{cr},\mathrm{gs},N_{e},\mathrm{fd}\right)}\left(\phi \right)\right|$$

$F_{\bar{\Phi }\left(\Lambda ,\eta ,\mathrm{ch},\mathrm{cr},\mathrm{gs},N_{e},\mathrm{fd}\right)}\left(\phi \right)$ denotes the cumulative density function (CDF) of the fatigue index values from the vector $\bar{\Phi }\left(\Lambda ,\eta ,\mathrm{ch},\mathrm{cr},\mathrm{gs},N_{e},\mathrm{fd}\right)$.

A third evaluation criterion (denoted C) aims at the disturbances of fatigue indexes. For all combinations of fatigue algorithms and EMG levels, a linear regression is performed for the fatigue index signal of the 60% MVC load case. The coefficient of determination $R^{2}$ is used to quantify how well the fatigue index signal resembles a linear slope. Deviations from the linear slope found in the 60% MVC load case are counted towards disturbances of the fatigue index. Thus, a smaller $R^{2}$ corresponds to a higher level of disturbances. Remaining cardiogenic artifacts must not result in a more rugged fatigue index. Therefore, it is desired that $R^{2}$ remains as close as possible to the reference case. Disturbances are quantified by: (10)$$\gamma_{C}\left(\eta ,\mathrm{ch},\mathrm{cr},\mathrm{gs},N_{e},\mathrm{fd}\right)=R^{2}\left({\hat{\Phi }}_{\Lambda ,\eta ,\mathrm{ch}}^{\mathrm{cr},\mathrm{gs},N_{e},\mathrm{fd}}\left(t\right){|}_{0\;s<t<60\;s}\right)$$

### 2.5. Statistical Analysis

In the following results section, different aspects are presented, such as the effects of the artificial ECG channel, a comparison of ECG artifact removal methods and fatigue methods. In some cases, two similar variants are shown (MNF and SMR5 fatigue detection algorithm or TSW and TSWD cardiogenic artifact removal) in columns next to each other. Each column consists of three diagrams reflecting the three evaluation criteria $\gamma_{A}$, $\gamma_{B}$ and $\gamma_{C}$. Within each graph, the comparison is grouped by EMG level η, as shown on the horizontal axis. In some cases, the pure sEMG case without any cardiogenic artifacts is shown in addition on the right (in lighter/special color). Each combination of ECG removal or fatigue detection algorithm variant, evaluation criteria, EMG level and comparison object (color coded items juxtaposed next to each other in the subsequent plots) is based on nine results from the nine subjects under consideration. The results are shown either as the arithmetic mean value over the nine subjects or as a violin plot reflecting the distribution across the nine subjects.

In some cases, the significance is provided as *p*-values. The *p*-values are calculated independently for each combination of variant, evaluation criteria and EMG level. This leads to group-wise comparisons of three or four groups with 9 values in each group (one for each subject). A Friedman test [39] was utilized to check whether there is at least one group significantly different from another. Based on the sum of ranks, the best performing group was selected. Regarding $\gamma_{A}$, smaller values are considered to be better, while for $\gamma_{B}$ and $\gamma_{C}$, larger values are used. A significance level of 0.05 and the Holm adjustment method according to [40] were applied to compare the best performing group to all other groups (1 to N comparison). If significance is found, it is noted in the figure.

## 3. Results

The following section starts with the presentation of results regarding the quality of the artificial test signals as well as the influence of EMG level and ECG channel. This is followed by results comparing the different EGC removal methods. After that, results concerning the fatigue detection algorithms are presented. This part is subdivided into a presentation of detailed parameters for each of the algorithms followed by a comparison of fatigue detection algorithms.

### 3.1. Quality of Artificial Signals

Characteristic properties of the test signals were evaluated in a first step to analyze the heartbeat irregularity. The minimum value during QRS, the QRS magnitude (peak-to-peak) and the maximum during the T wave were gauged for each heart cycle. As this evaluation aimed at the ECG component only, it was performed based on ${\mathrm{ECG}}_{I}\left(t\right)$ and ${\mathrm{ECG}}_{A}\left(t\right)$ instead of the artificial test signals ${\mathrm{ATS}}_{\Lambda ,\eta ,\mathrm{ch}}\left(t\right)$, additionally including EMG with different magnitudes. The variance was slightly elevated for some subjects and characteristic properties. However, no distinctive difference could be found comparing the ECG channel I with the artificially mixed ECG channel. A visualization of the characteristic properties can be found in the Appendix B Figure A2.

An alternative method to quantify the heartbeat irregularity is based on the fraction of ECG artifacts that could not be eliminated by template subtraction from the signals ${\mathrm{ATS}}_{60,\eta ,\mathrm{ch}}\left(t\right)$ with EMG levels η=0.01,0.02,0.05and0.1. Regarding the fraction of residual QRS artifacts that could not be eliminated by template subtraction we found a distinctive difference. While the fraction of QRS artifacts that could not be eliminated by template subtraction is on average at 3.25% in the case of ECG channel I, it is elevated at 7.29% for the artificially mixed ECG channel A with increased heartbeat irregularity. Further details about the quantification of heartbeat irregularity can be found in the Appendix B Figure A2.

### 3.2. EMG Level η and Artificially Mixed ECG Channel

Regarding the different EMG levels, the fatigue detection unsurprisingly performs the worst at low EMG levels η featuring high ECG contamination and only a small share of desired sEMG. At higher EMG levels η, the fatigue detection results become more and more equivalent to the uncontaminated cases. The fatigue detection based on ${\mathrm{ECG}}_{I}$ shows better results compared to those from the artificially mixed ${\mathrm{ECG}}_{A}$. This effect is again most distinctive at low EMG levels η. A graphical representation of the results from different ECG channels at various EMG levels can be found in Figure 4.

The evaluation of the uncontaminated cases allows for an estimation of the impact of the breathing phase modulation. While both uncontaminated cases omit artificially added cardiogenic artifacts, the first ($\Xi_{\Lambda }^{\mathrm{GN},256,\mathrm{fd}}\left(t\right)$) features a breathing phase modulation, while the later ($\Psi_{\Lambda }^{\mathrm{GN},256,\mathrm{fd}}\left(t\right)$) does not and is based on the pure and unmodified sEMG signal. Comparing both uncontaminated cases against each other, hardly any difference could be observed. The results are also visualized in Figure 4.

### 3.3. Performance of ECG Removal Algorithms

Different approaches to deal with the cardiogenic artifacts were compared, including the sole high-pass filter (HP15), template subtraction with and without wavelet denoised template (TS15 and TSW15), template subtraction with wavelet denoised template and subsequent damping step (TSWD15) and the application of the damping step without template subtraction (DSO). No treatment of the cardiogenic artifacts is included for reference. Measures quantifying the deviation of the artifact-reduced signals compared to the pure EMG component in the time and frequency domain were investigated in a first intermediate step. Regarding the ECG “cleaned” electromyographic signals in the time and frequency domain, the best results were obtained with the TSDW15 and DSO methods. In the case of envelope calculation based on the ECG signal, TSWD15 yielded best results. More detailed results can be found in Figure A9 in the Appendix B.

However, the main focus of this study is on the influence of ECG removal on the subsequent fatigue detection. Figure 5 shows the influence on MNF and SMR5 fatigue detection caused by different approaches dealing with the cardiogenic artifacts. The exclusion of QRS segments (gating) is checked for each method additionally. For better clarity, only the average across all subjects is considered for each ECG removal method.

Comparing the high-pass filter (HP15) to the non-compensation case, if any, only slight improvements could be observed across both MNF and SMR5 fatigue detection methods and the three evaluation criteria. Both template subtraction methods (TS15 and TSW15) considerably improved the fatigue detection. The utilization of the wavelet denoising step during the template construction (TSW15) caused only tiny improvements for both MNF and SMR5. However, even regarding template subtraction TSW15, gating is still delivering better results except at very high EMG levels η. The introduction of the wavelet-based damping step downstream to the template subtraction (TSWD15) leads to a further considerable improvement, especially in the case of low and medium EMG levels η. In contrast to the other applied methods, TSWD15 reduced the cardiogenic artifacts so well that it is reasonable not to apply gating and instead include the QRS segments also in fatigue detection. Outcomes of the sole damping step (DSO) are behind those of TSWD15.

Figure 5 also visualizes how the reference fatigue index signal $\Xi_{\Lambda }^{\mathrm{GN},256,\mathrm{fd}}\left(t\right)$ is impaired if gating is applied ($\Xi_{\Lambda }^{\mathrm{GO},256,\mathrm{fd}}\left(t\right)$). Note that only the position of the gating is taken from a contaminated signal. Both signals contain no cardiogenic artifacts. The sole interruptions due to gating already impose a noticeable drop in fatigue detection quality.

Figure 6 visualizes a more detailed comparison of a selection of ECG removal algorithms in combination with SMR5 fatigue detection. This figure not only displays the average values but visualizes the underlying distribution across subjects as a violin plot. Furthermore, it states *p*-values if significant differences were found. In this detailed comparison, TSWD15 scored best in all cases. It always scored significantly better than HP15. In the case of small EMG levels η, it performed significantly better than TSW15, and in the case of larger EMG levels, there was a significant improvement compared to DSO. Further, a detailed visualization for the best performing methods can be found in the Appendix B
Figure A10 and Figure A11).

### 3.4. Detail Parameters for PSD-Based Fatigue Algorithms

For both MNF and SMR5, no evidence could be obtained that larger epoch sizes (e.g., $N_{e}=512\;\mathrm{and}\;1024$) lead to better results. Fatigue signals based on a moving average filter combining fatigue values from smaller segments within a 1 s window showed similar or better results. In the case of the mean fatigue detection (MNF), larger epoch sizes even tend to worsen results. A visualization of results based on different epoch sizes $N_{e}$ moving average filtered to fatigue index values incorporating 1 s of EMG can be found in the Appendix B Figure A12.

Regarding MNF fatigue detection, the non-parametric Welch PSD estimation yielded superior results compared to PSD estimation with the Burg method. However, the optimal choice of the number of subsegments depends on the EMG level. At lower EMG levels η, the utilization of $k_{w}=15$ tends to yield good results, while at higher EMG levels, $k_{w}=31$ were found to be better. The SMR5 fatigue detection algorithm quite constantly displayed good to very good results in connection with the parametric Burg PSD estimation. A visualization can be found in the Appendix B Figure A12).

An evaluation of the influence of the PSD lower bound $b_{l}$ can also be found in the Appendix B Figure A13.
While the optimal $b_{l}$ is in the range of 35 to 40 Hz for low EMG levels η, it is lower (towards 20 Hz) for higher EMG levels.

### 3.5. Detail Parameters for fApEn

In the case of fApEn, the exclusion of lower frequency components is realized by a high-pass filter. The ECG “cleaned” signals are alternatively filtered with 35 Hz corner frequency. Especially at low EMG levels, a high-pass filter with 35 Hz corner frequency showed better results. Akin to the PSD-based fatigue detection method, it positively effects separability ($\gamma_{B}$) and disturbances ($\gamma_{C}$) at smaller EMG levels η. However, comparing both corner frequencies, we observed an increased discrepancy to the reference fatigue index $\Phi \left(t\right)$ reflected by $\gamma_{A}$. This shift does not matter in practice as the evaluation criteria, separability (B) and disturbances (C), are hardly impaired. In the case of higher EMG levels, both the 15 Hz and the 35 Hz variants supply similar results. Larger epoch sizes yielded slightly poorer results regarding fatigue detection by fApEn. This applies to epoch sizes $N_{e}=512\;\mathrm{and}\;1024$ compared to those with 256 samples. The raw detection results were filtered with a moving average filter incorporating 1 s of sEMG data regardless of epoch size to assure comparable settings. In contrast to the PSD-based fatigue detection methods, QRS gating had no positive effect for TSWD even at very low EMG levels. However, in the case of gating, the epoch size of 128 samples yielded worse results. A visualization of results based on different epoch sizes $N_{e}$ (moving-average-filtered to assure that fatigue index values incorporate 1 s of sEMG) can be found in the Appendix B Figure A11a.

### 3.6. Fatigue Algorithms

A final comparison, including the most promising variants of MNF, SMR5 and fApEn, is shown in Figure 7 and Figure 8. The deviation to the noncontaminated case measured by $\gamma_{A}$ is comparable for both MNF and SMR5 algorithms if TSWD is used to eliminate cardiogenic artifacts. However, if TSW is used in combination with gating (GO), SMR5 performed better than MNF regarding criterion A. $\gamma_{A}$ is considerably increased for fApEn. Regarding the separation of 20% and 60% MVC load cases (criterion B), the SMR5 algorithm yielded the best results. In most cases, SMR5 is significantly better than MNF and fApEn for $\gamma_{B}$. Corresponding *p*-values are stated in Figure 8. The disturbances (criterion C) were also the lowest for SMR5 in many cases. In the case of high EMG levels, fApEn showed slightly higher $\gamma_{C}$ values in the case of TSW and gating. However, fApEn shows the disadvantage of inter-subject variability (Note the size of the violins in Figure 8). MNF results are inferior to those of SMR5 with respect to $\gamma_{C}$ (criterion C).

## 4. Discussion

The goal of this study is to identify the most reasonable signal processing workflow to monitor muscle fatigue even in sEMG signals heavily influenced by cardiogenic artifacts. The main focus of this work was on two key components of the signal processing and their combination: (1) elimination of cardiogenic artifacts and (2) fatigue detection algorithms playing well together with the elimination methods.

The results underline the importance of eliminating cardiogenic artifacts from respiratory sEMG signals prior to fatigue detection. The application of template subtraction (TS) considerably improved the fatigue detection results compared to the sole application of a high-pass (HP15) or no treatment at all. Both sub-variants with and without wavelet denoising of the template (TSW and TS, respectively) fall very close to each other. It was demonstrated that, even in the case of template subtraction (TS and TSW), excluding QRS segments by gating causes considerable improvement. This effect is especially present at small or medium EMG levels η. It leads to the conclusion that neither HP15, TS nor TSW alone are able to eliminate QRS artifacts sufficiently enough to abandon gating. In practice, this implies that the cardiogenic artifacts are still so pronounced in the QRS segments that it is better to disregard them. We indeed demonstrated by the help of the uncontaminated reference signals $\Xi \left(t\right)$ that gating itself changes the overall fatigue results only to some degree. Nevertheless, the necessity of gating highlights that there is still room for improvement.

One of such improvements is introduced by the additional wavelet damping step downstream to the template subtraction (TSWD). The remaining cardiogenic artifacts during the QRS segment are, in most cases, reduced thus far as to make sense to also include these segments into fatigue detection algorithms. The method successfully considers the variations of the shape of the cardiogenic artifact. An alternative approach by Deng et al. [33] introduced an adaptive gain to scale the template and find a best fit by minimizing the squared error. However, it is questionable whether the shape change of the cardiogenic artifact consists of a pure scaling only. Instead of this, it can be assumed that the shape changes in a more complicated way. Our TSWD method is able to cope with these more complex shape changes. The exclusive application of a wavelet damping without template subtraction (DSO) produced comparable error measures in the intermediate analysis of the electromyographic signals. However, the fatigue detection results based on DSO were poor especially at higher EMG levels. Only the combination of the additive template subtraction and a downstream wavelet-based multiplicative damping step in the method TSWD arranged for a reasonable elimination of cardiogenic artifacts under increased heartbeat irregularity in view of a subsequent fatigue detection. This observation indicates that the damping step can only play out its strength when applied subsequently to ordinary template subtraction. Apparently, TSWD combines the advantage of sufficient template subtraction reducing cardiogenic artifacts at high EMG levels, and those of the damping step being better suitable at smaller EMG levels. The fact that the intermediate results of DSO looked promising in contrast to the fatigue detection results underlines the importance of this evaluation. When it comes to sEMG-based fatigue detection, it is not sufficient to select an ECG removal algorithm based on error measures from the resulting electromyographic signal. The fatigue detection seems to rely on features in the signal that are not adequately covered by these error measures.

The best fatigue detection results were found with epoch sizes of 128 and 256 samples. However, the results based on epoch size $N_{e}=128$ partially displayed a tendency to fluctuate results, especially in connection with fApEn and gating. This effect is probably related to the length of the epoch (0.125 s) being too similar to the length of gating (0.15 s). This favors the formation of small fragments, which might mislead fatigue detection. Thus, it seemed most appropriate to use an epoch size $N_{e}=256$ for the basic comparison. It was shown that larger epoch sizes demonstrate no advantage beyond an averaging effect. If the results of smaller epoch sizes were filtered with a moving average filter to achieve a similar averaging effect, the results were similar or even better. A possible explanation for the advantage of smaller epoch is that residual cardiogenic artifacts are usually concentrated at certain heartbeat phases. With small epoch sizes, they can only distract one of many small epochs in a signal segment. This single impaired epoch has only a limited influence if many epochs are averaged. Furthermore, the computational complexity is often lower in the case of shorter epoch sizes.

The cardiogenic artifacts tend to have lower frequency components than sEMG signals. However, the frequency ranges are strongly overlapping [32]. Nevertheless, it is useful to exclude lower frequencies from the fatigue detection. This is reflected by the lower bound $b_{l}$ for PSD-based algorithms MNF and SMR5 and the choice of the high-pass filter corner frequency in the case of fApEn. The choice of $b_{l}$ is a trade-off between cutting of the necessary lower bands with intense residual artifacts while keeping the relevant EMG components at higher frequencies. There is no single choice of $b_{l}$ that is optimal in all situations and across different performance evaluations. Reflecting the idea that detection is most crucial at lower EMG levels η, we conclude that, in general, $b_{l}=35\;\mathrm{Hz}$ is favorable.

Among the analyzed fatigue detection methods, SMR5 showed best results. This algorithm was the best to cope with cardiogenic artifacts. Regarding the separation of different fatigue levels ($\gamma_{B}$), it scored significantly better than mean frequency (MNF) and fuzzy approximate entropy (fApEn) in all but the lowest EMG level. The idea behind SMR5 to consider high frequencies fits together very well with the fact that cardiogenic interferences are mainly present in lower frequencies. Outside the QRS, cardiogenic artifacts are concentrated in lower frequencies. As a result, SMR5 delivers good results even with suboptimal ECG removal if gating is applied. This can be seen from the fact that, in Figure 5, the dotted lines are close together for SMR5 (right column) in contrast to MNF (left column). Even if the treatment of artifacts consists of gating only, the SMR5 algorithm can already detect fatigue quite well, compared to other fatigue algorithms.

The intra-heartbeat fatigue detection scheme proposed by Sinderby et al. [10] cannot be directly compared due to different epoch sizes. It was suggested to select the epoch based on the heart cycle, leading to epoch sizes of a quarter of the heart cycle length. A comparison based on signals to include 1 s long epochs can be found in the Appendix B Figure A14. In most combinations of EMG levels and evaluation criteria, our artifact removal TSW15 with gating strategy GO and TSWD15 with no gating scored better than this approach. In many cases, and especially regarding separability $\gamma_{C}$ and low EMG levels η, the intra-heartbeat approach was significantly worse. Unfortunately, it was not possible for us to include the RLS approach suggested by Ortega et al. [11] into the comparison.

The analysis is based on artificially constructed sEMG signals. These signals were not intended to be a realistic copy of real-life signals. Instead, their purpose was limited to cover particular effects such as increased heartbeat irregularity and a clearly known expected fatigue level. The signals were designed with a focus on the repetitive cardiogenic artifacts. Other effects such as non-repetitive high-frequency components of the ECG were not included. Neither was the focus put on the specifics of sEMG from cyclic contractions. Comparing both uncontaminated cases ($\Xi_{\Lambda }^{\mathrm{GN},256,\mathrm{fd}}\left(t\right)$ and $\Psi_{\Lambda }^{\mathrm{GN},256,\mathrm{fd}}\left(t\right)$ in Figure 4) against each other leads to the conclusion that the sEMG modulation by simulated breathing on its own causes hardly any effect.

The performance of template subtraction highly depends on whether basic theoretical assumptions are met [33]. Deng et al. [33] mentions this issue as an explanation of why results based on simulated data sometimes look very promising, while trials with real-life data display poorer results. Among other things, the basic assumptions of the template subtraction include the regularity of the repetitive ECG artifact. With the introduction of an artificially mixed ECG channel, we tried to increase the heartbeat irregularity so that our signals are more realistic in this respect. By comparing the artificial ECG signal with ECG channel I, it could be shown that this goal was achieved. While characteristic measures, such as minimum value during QRS, QRS magnitude and T-wave maximum, changed only slightly, the fraction of residual QRS artifacts irremovable by template subtraction more than doubled. Furthermore, the fatigue detection results based on ECG channel I yield better results compared to those based on the artificially mixed ECG channel. This reflects the desired property of the artificially mixed ECG channel to feature a higher heartbeat irregularity. The increased heartbeat irregularity leads to a less effective artifact removal by template subtraction and thus to worse fatigue detection results.

Artificially constructed test signals were also used for the evaluation of algorithms eliminating cardiogenic artifacts from sEMG signals by Deng et al. [33] and Petersen et al. [15]. Our approach differs in two key aspects. First, we constructed the artificially mixed ECG channel to increase the heartbeat irregularity within each test signal. We did not emphasize a completely realistic breathing phase modulation of ECG. Important for our analysis was the heartbeat irregularity and the burden on a subsequent template subtraction. We wanted to assure that the effect of the QRS shape changing correlated with the filling level of lungs is included. In contrast to the procedure used by Deng et al. [33], our approach assures a shape variation beyond a pure scaling of the heart segments by a random gain factor. Secondly, we used an artificially modulated sEMG signal from a static contraction to mimic the cyclic breathing activity. This way, we could assure a clearly defined expected result, as the underlying pure sEMG signal from static contractions offered a linear onset in fatigue measures. Our approach to modulate an sEMG signal from a static contraction can be seen as a compromise between approaches used by Deng et al. [33] and Petersen et al. [15]. The first group of authors used differently filtered versions of white Gaussian noise for inspiratory and expiratory segments. The second group used periodic and irregular sEMG recordings from musculus gastrocnemius. For our purpose, it was important to have this clearly defined expected fatigue result as we wanted to evaluate the resulting fatigue detection signal and not the “cleaned” sEMG like both studies mentioned above.

As already mentioned, the artificial test signals are based on assumptions regarding heartbeat irregularity. These assumptions almost certainly have an influence on how large the non-removable residual artifacts are and thus how severely the fatigue index is disturbed. To reflect this issue, we discussed most aspects for two different settings. The first setting is based on the TSW removal algorithm with the gating strategy GO, and the second setting is based on TSWD with no gating. The results for the first setting would also be valid to a large degree if the assumptions are not met because the residual artifacts are for the most part within the QRS segment that is eliminated by gating anyway. The second setting covers the best opportunity if the assumptions are met. It is also relevant in cases where the heartbeat irregularity is less pronounced in real life signals than in our test signals. It can be concluded that the design of test signals is a trade-off between how close the test signals are to real respiratory EMG signals and how easy they can be evaluated. Overall, we are convinced that the utilized signals are a reasonable choice.

Our analysis focused on the ability of fatigue algorithms to cope with residual cardiogenic artifacts arising from regular heart activity. In our analysis, these residual artifacts are leftovers from improper attenuation of deterministic ECG components. Further studies might investigate the influence of other effects present in respiratory sEMG recordings. Among them are, for example, EMG interferences from other muscles, movement artifacts or other disturbances. It also requires the study of non-ordinary cardiac activity on the performance of artifact suppression and fatigue detection. This includes, e.g., deviations from normal sinus rhythm such as arrhythmic or premature heartbeats.

## 5. Conclusions

The results illustrate that the selection of both the cardiogenic artifact suppression algorithm as well as the actual fatigue detection algorithm play a crucial role when it comes to detecting fatigue from sEMG signals with cardiogenic artifacts. The newly introduced two-step variant of template subtraction (TSWD) scored best at suppressing cardiogenic artifacts. In many situations, TSWD is the only method where it seems reasonable to eliminate gating. Regarding the fatigue detection algorithms, the spectral moments ratio (SMR5) turned out to be most suitable for signals with previously removed cardiogenic artifacts. The focus of SMR5 on high frequencies, which are hardly affected by cardiogenic artifacts, is particularly useful here. Consequently, the combination of TSWD and SMR5 can be recommended for fatigue detection from respiratory sEMG. This work provides guidance for clinical studies to select optimal signal processing to detect fatigue from respiratory sEMG signals. This seems especially important since the algorithms from existing clinical trials are considerably underperforming [10] or not sufficiently described to be reproduced [11] in our test scenarios. When the determination of fatigue from respiratory sEMG signals becomes robust and reliable, it will become an interesting tool in clinical decisions.

## Figures and Tables

**Figure 1 sensors-21-05663-f001:**
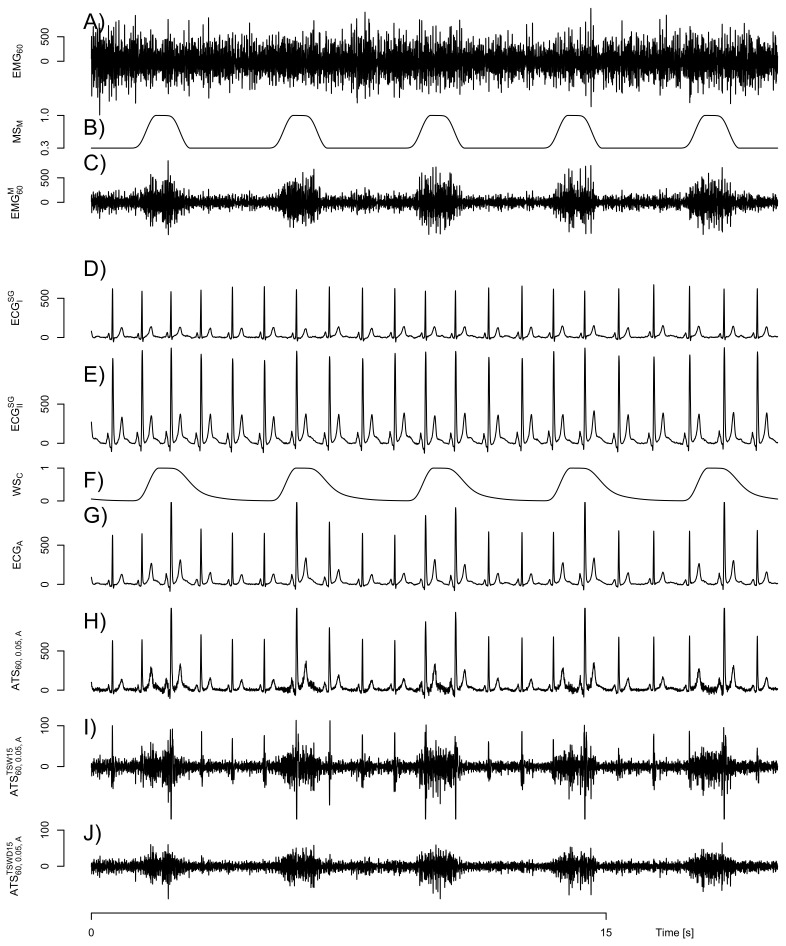
Artificial signal construction based on electrocardiographic (ECG) and surface electromyographic (sEMG) signals. (**A**) shows an sEMG signal from a 60% MVC load case (${\mathrm{EMG}}_{60}\left(t\right)$) taken from [16]. Modulation with the weighting signal ${\mathrm{MS}}_{M}\left(t\right)$ (displayed in (**B**)) to mimic a breathing-like activity pattern yields ${\mathrm{EMG}}_{60}^{M}\left(t\right)$ displayed in (**C**). (**D**,**E**) display ECG signals ${\mathrm{ECG}}_{I}^{\mathrm{SG}}\left(t\right)$ and ${\mathrm{ECG}}_{\mathrm{II}}^{\mathrm{SG}}\left(t\right)$ from channel I and II, respectively. (**F**) shows the weighting signal ${\mathrm{WS}}_{C}\left(t\right)$ roughly corresponding to the lung volume utilized to mix both ECG channels. Its application yields ${\mathrm{ECG}}_{A}\left(t\right)$ with an increased irregularity displayed in (**G**). The artificial test signal ${\mathrm{ATS}}_{60,0.05,A}\left(t\right)$ based on ${\mathrm{ECG}}_{A}\left(t\right)$ and ${\mathrm{EMG}}_{60}^{M}\left(t\right)$ is drawn in (**H**). Note the small amplitude of sEMG components. (**I**) shows a “cleaned” signal ${\mathrm{ATS}}_{60,0.05,A}^{\mathrm{TSW}15}\left(t\right)$ with cardiogenic artifacts removed by template subtraction. Note the change in scale as compared to (**H**). Some cardiogenic artifacts are still clearly visible in the signal due to the intended ECG irregularity of the artificial signal. (**J**) shows a signal ${\mathrm{ATS}}_{60,0.05,A}^{\mathrm{TSWD}15}\left(t\right)$ with further reduced residual cardiogenic artifacts. All physiological signals are displayed in μV.

**Figure 2 sensors-21-05663-f002:**
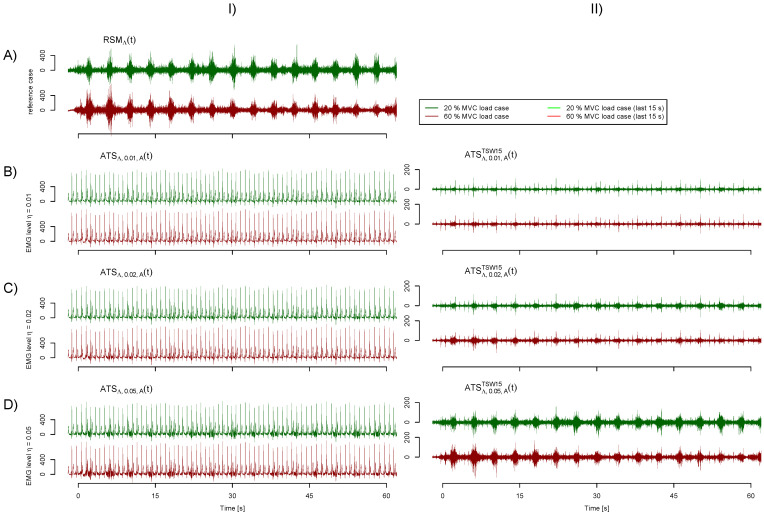
Different EMG levels mixed with ECG. The first main row (**A**) depicts the reference case, including the sEMG modulation but no ECG contamination. The following main rows (**B**–**D**) depict the cases with η=0.01,0.02and0.05, respectively. For each situation, the 20% and 60% maximum voluntary contraction (MVC) load cases are shown in the first and second sub row. Column (**I**) shows the reference signal ${\mathrm{RSM}}_{\Lambda }\left(t\right)$ in the case of row (**A**) and artificial test signals ${\mathrm{ATS}}_{\Lambda ,\eta ,A}\left(t\right)$ otherwise. The signals with cardiogenic artifacts removed by template subtraction ${\mathrm{ATS}}_{\Lambda ,\eta ,A}^{\mathrm{TSW}15}\left(t\right)$ are shown in the column (**II**). Corresponding fatigue index signals and their evaluation is visualized in Figure 3.

**Figure 3 sensors-21-05663-f003:**
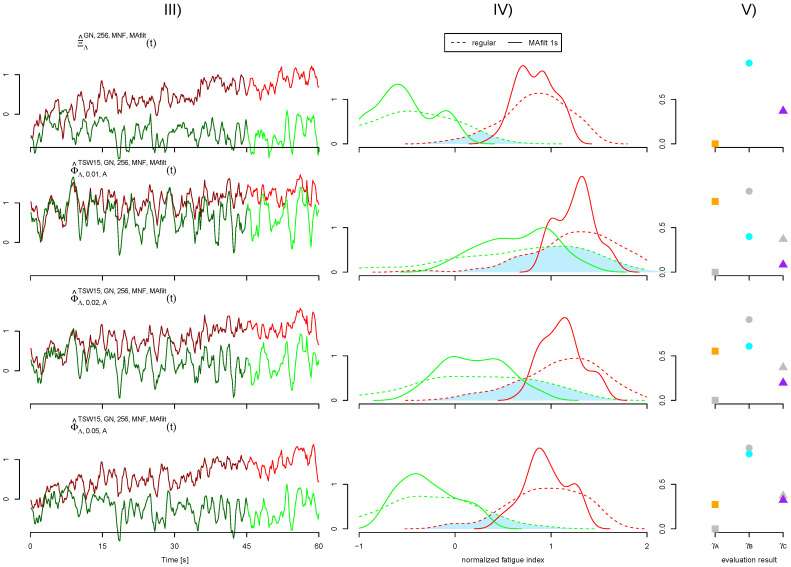
Exemplary fatigue index signals and their evaluation corresponding to exemplary test signals depicted in Figure 2. The first row represents the reference case, while the following rows visualize cases with η=0.01,0.02and0.05, respectively. Column (**III**) shows the resulting filtered fatigue index signals for the 20% and 60% MVC load cases in green and red, respectively. Note that each filtered fatigue index data point incorporates sEMG from a 1s long window. The mean frequency (MNF) is used as fatigue algorithm with an epoch size of $N_{e}=256$, power spectral density (PSD) estimation with the Welch method ($k_{w}=15$) and a lower bound $b_{l}=35\;\mathrm{Hz}$. Column (**IV**) shows the distribution of values $\bar{\Phi }$ arising from the last 15 s of both load cases. The difference to one of the overlapping area of both distributions (light blue) corresponds to the Kolmugorov–Smirnov distance used to quantify the separability of fatiguing and non-fatigued situations by $\gamma_{B}$. The resulting performance evaluation values $\gamma_{A}$, $\gamma_{B}$ and $\gamma_{C}$ for deviation to the reference fatigue index (■), separability of load cases (●) and disturbances measured by $R^{2}$ (▲), respectively, are shown in column (**V**). Values arising from the reference case are drawn in light gray for all other EMG levels.

**Figure 4 sensors-21-05663-f004:**
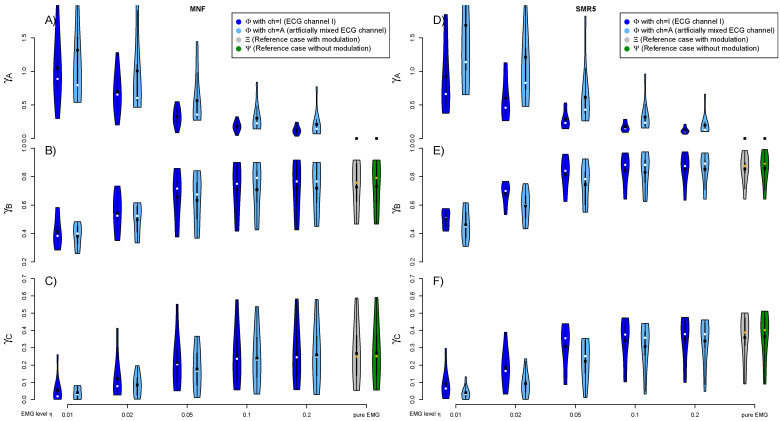
Fatigue detection results depend on the magnitude and characteristics of the cardiogenic artifacts. This figure juxtaposes the results originating from ECG channel I ($\Phi_{\Lambda ,\eta ,I}^{\mathrm{TSW}15,\mathrm{GN},256,\mathrm{fd}}\left(t\right)$) and the artificially mixed ECG channel A ($\Phi_{\Lambda ,\eta ,A}^{\mathrm{TSW}15,\mathrm{GN},256,\mathrm{fd}}\left(t\right)$). Each subfigure includes different EMG levels η plotted on the horizontal axis. Additionally, two cases without ECG contamination are included for reference. The first uncontaminated case contains breathing phase modulation $\Xi_{\Lambda }^{\mathrm{GN},256,\mathrm{fd}}\left(t\right)$ (based on ${\mathrm{RSM}}_{\Lambda }\left(t\right)$), while the later $\Psi_{\Lambda }^{\mathrm{GN},256,\mathrm{fd}}\left(t\right)$ (based on ${\mathrm{RS}}_{\Lambda }\left(t\right)$) does not. Each row resembles one of the applied evaluation criteria. The discrepancy of fatigue indexes originating from the contaminated signals in relation to the reference fatigue index with modulated EMG only (quantified by $\gamma_{A}$) is shown in the first row (**A**,**D**). The smaller the deviation, the better the fatigue algorithm’s performance. The second row (**B**,**E**) visualizes the separability of the 20% and 60% MVC load cases (quantified by $\gamma_{B}$, the higher the better). The level of disturbances measured by the coefficient of determination $R^{2}$ (quantified by $\gamma_{C}$) is depicted in the last row (**C**,**F**). The higher $R^{2}$, the better the fatigue detection method. The template subtraction method TSW15 was applied to compensate cardiogenic artifacts. The subfigures on the left are based on MNF with Welch PSD estimation ($k_{w}=15$). The subfigures to the right are based on spectral moments ratio of order five (SMR5) with Burg PSD estimation. Both fatigue detection methods are based on an epoch size $N_{e}=256$ and utilized a PSD lower bound $b_{l}=15\;\mathrm{Hz}$. The black dots indicate the mean value, while the white/orange dots symbolize the median.

**Figure 5 sensors-21-05663-f005:**
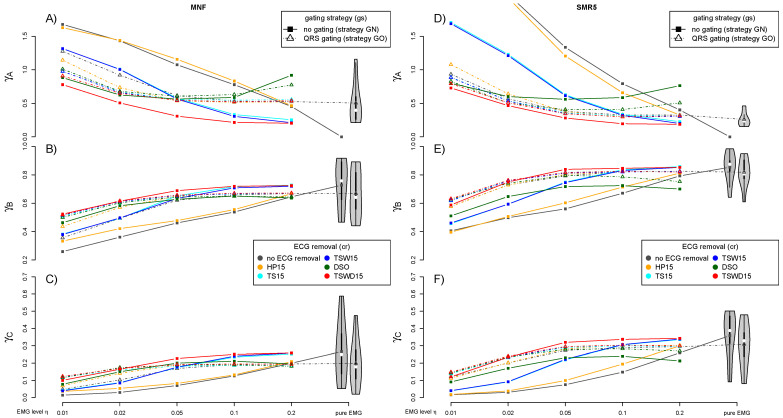
The effect of cardiogenic artifact removal on the subsequent fatigue detection. Different methods to reduce the cardiogenic artifacts are compared next to omitting any artifact treatment (gray). Included is the sole application of a high-pass filter (HP15 with $f_{c}=15\;\mathrm{Hz}$) and two different variants of template subtraction (TS15 and TSW15). The latter template subtraction utilizes the additional wavelet denoising of the template. Additionally shown is heartbeat synchronous damping in different wavelet scales applied solely (DSO) and subsequent to template subtraction (TSWD15). The results considering gating are shown for each method separately (dashed line-gating strategy GO). In addition to the different EMG levels η, each figure also includes two cases without any ECG contamination. The first case (gray filled) evaluates $\Xi_{\Lambda }^{\mathrm{GN},256,\mathrm{fd}}\left(t\right)$, while the second one evaluates $\Xi_{\Lambda }^{\mathrm{GO},256,\mathrm{fd}}\left(t\right)$. The second case (gray shaded) introduces a gating based on time information taken over from pure ECG signal. It shows the impact of causeless gating (despite cardiogenic artifacts lacking). This allows an assessment of how much the fatigue detection quality drops by the sole interruptions due to gating. The three applied evaluation criteria are again covered by the rows of subfigures ((**A**,**D**), (**B**,**E**) and (**C**,**F**), respectively), while MNF and SMR5 represent the columns. The Welch method ($k_{w}=15$) is utilized for MNF, while SMR5 relies on the Burg method. $N_{e}=256$ and $b_{l}=35\;\mathrm{Hz}$ are applied for both methods. Each data point is obtained by averaging over all subjects. A more detailed comparison for a selection of ECG removal algorithms in combination with SMR5 and no gating (corresponding to the solid lines of the right column) can be found in Figure 6. Further comparisons for the algorithms TSW15 and TSWD15 can be found in the Appendix B Figure A10 and Figure A11).

**Figure 6 sensors-21-05663-f006:**
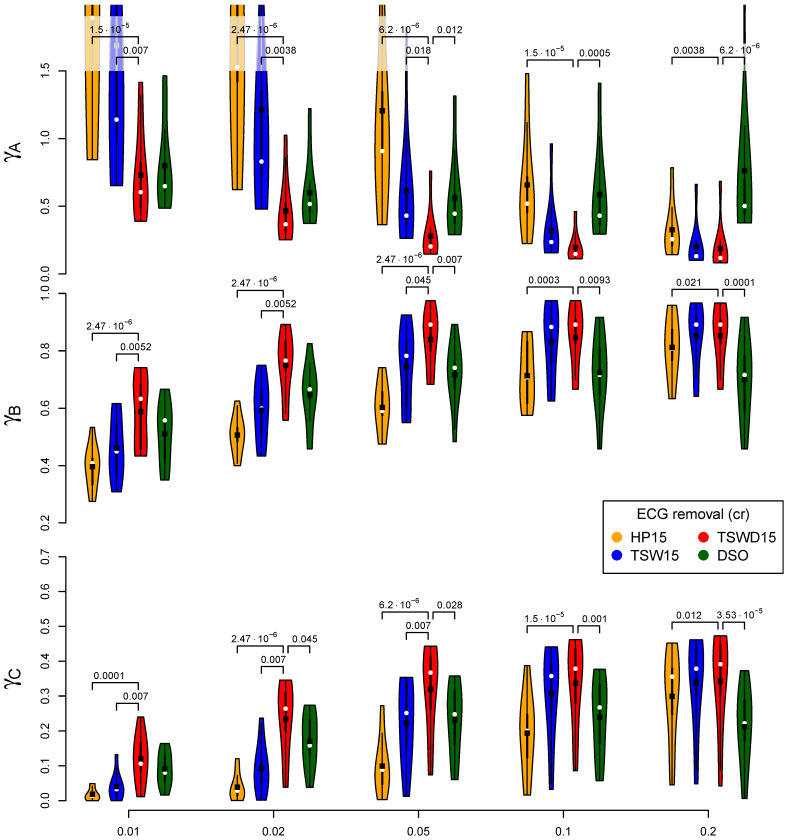
A detailed comparison of selected cardiogenic artifact removal methods regarding their impact on the subsequent fatigue detection by SMR5 method and no gating. The shown violin plots correspond to solid lines in the same color from the right column of Figure 5. The best performing ECG removal method was determined for each group of four violin plots sharing the same EMG level and evaluation category. The existence of significant differences to the other three algorithms were evaluated for each group. If significant differences exist, the corresponding p-values are stated. SMR5 is calculated with $N_{e}=256$ and $b_{l}=35\;\mathrm{Hz}$ and utilized the Burg method for spectral estimation.

**Figure 7 sensors-21-05663-f007:**
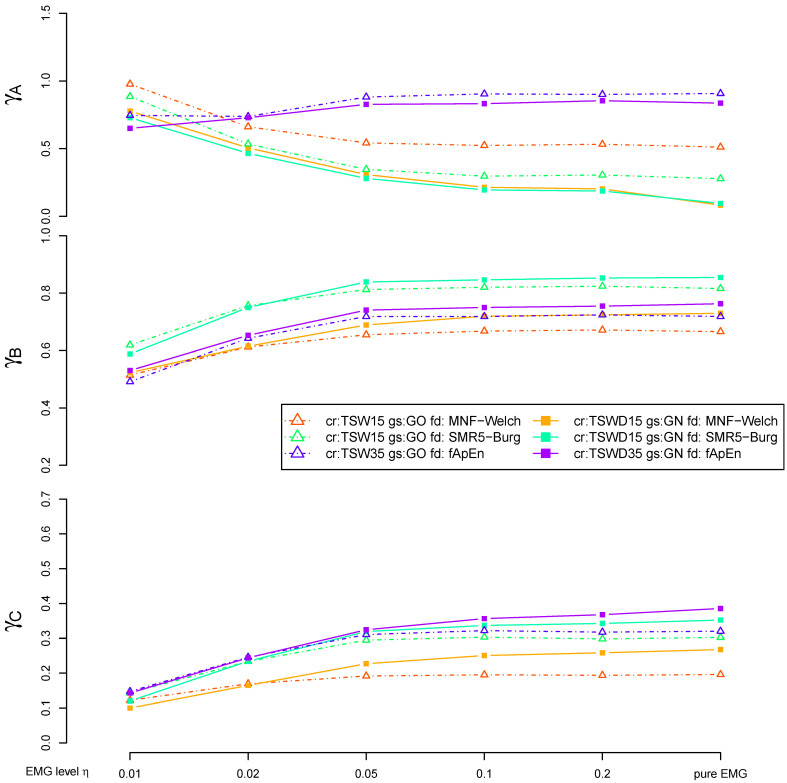
A comparison of fatigue detection algorithms. The results of MNF, SMR5 and fuzzy approximate entropy (fApEn) methods are shown in combination with ECG removal by TSWD and no gating (solid lines) as well as TSW and gating (dashed dotted lines). MNF is utilized with Welch ($k_{w}=15$) and SMR5 with Burg PSD. Both PSD-based algorithms include a PSD lower bound $b_{l}$ of 35 Hz, while fApEn includes a 35 Hz high-pass filter within the ECG removal step. All fatigue detection methods are based on epoch size $N_{e}=256$. The results are shown as the mean value obtained by averaging over all subjects. Note that the distributions underlying the shown mean values are visualized in Figure 8.

**Figure 8 sensors-21-05663-f008:**
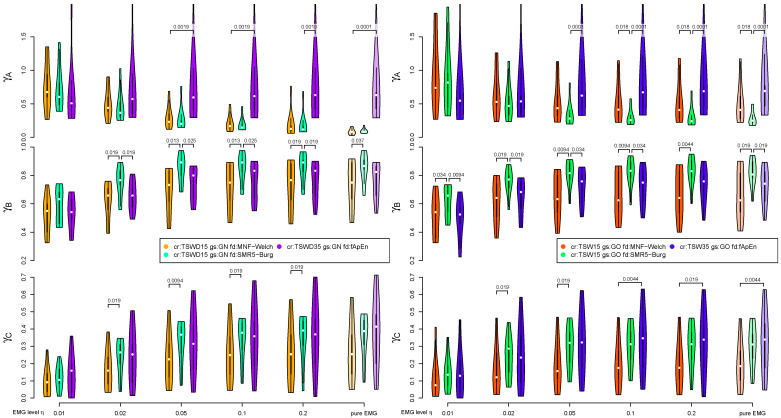
A detailed comparison of fatigue detection algorithms. The left column displays the results based on TSWD artifact removal method (including the wavelet-based damping step) without QRS gating (corresponding to solid lines in Figure 7). The right visualizes the results based on template subtraction TSW15 with QRS gating (GO), corresponding to dashed dotted lines in Figure 7. Fatigue algorithms include MNF, SMR5 and fApEn with the same parameters as in Figure 7. The best-performing fatigue detection method was determined for each group of three violin plots sharing the same EMG level and evaluation category. The existence of significant differences to the other two methods were evaluated for each group. If significant differences were found, the corresponding p-values are stated. The white dots symbolize the median value. The horizontal axis denotes EMG levels η. The three applied evaluation criteria (A, B, C) are covered by the rows.

## Data Availability

The test signals presented in this study are available in the Appendix A. This includes signals ${\mathrm{ATS}}_{\Lambda ,\eta ,\mathrm{ch}}\left(t\right)$, ${\mathrm{RSM}}_{\Lambda }\left(t\right)$ and ${\mathrm{RS}}_{\Lambda }\left(t\right)$.

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
