# Peer review of "Removal of ECG Artifacts Affects Respiratory Muscle Fatigue Detection—A Simulation Study"

_sensors, 2021, doi:10.3390/s21165663_

Round 1

Reviewer 1 Report

I have only a brief comment. In the Discussion section, please remove the text paragraph located between lines 652 and 657. It gives no merit to the current work to disparage against a work that was already published in literature (ref. [11]).

Otherwise, congrats for the proposed work on the detection of fatigue in sEMG recordings. It remains to be seen if the algorithm proposed by the authors can be implemented in real-time with the simultaneous acquisition of bio-potentials, and not rely exclusively in off-line computations. 

Author Response

Thank you for the review.

Except from the first sentence "Unfortunately, it was not possible for us to include the RLS approach suggested by Ortega et al. [11] into the comparison.", we deleted the paragraph between lines 652 and 657.

This remaining sentence (that should not be disparaging at all) is kept, because there is also a reference to that paper [11] in the conclusion, that has been slightly adopted:

"This seems especially important since the algorithms from existing clinical trials are considerably underperforming [10] or not sufficiently described to reproduce [11] in our test scenarios"

Reviewer 2 Report

Thank you for improving description of results through a better use of figures.

I understand that the clinical trials will be subject of a follow-up paper and I think the paper can be published in its present form.
Of course including a few exemplary data from the ongoing experiments would have increased confidence in the proposed method, but that is definitely not mandatory.

Author Response

Thank you for the review.

This manuscript is a resubmission of an earlier submission. The following is a list of the peer review reports and author responses from that submission.

Round 1

Reviewer 1 Report

Kahl et al. compared different methods to eliminate artifacts due to cardiac activity from sEMG signals and evaluated the impact of residual artifacts on fatigue detection. The method is very interesting although testing  only on simulated signals represents a limitation. Presentation needs to be improved, especially in the results.

Specific comments:

  • The sEMG signals description is presented after the ECG contamination description. I suggest reviewing this paragraph in order to make the data description clearer.
  • Authors used the lead I and II as contaminating signals in order to simulate the cardiac interference on respiratory muscles electrical activity. I am concerned about the choice of these two peripheral leads instead of precordial leads. Can the authors discuss this choice?
  • The term “cardiogenic” is not completely clear. Moreover, the authors displayed only ECG with sinus rhythm, in Figure 1. I suggest discussing the possibility to have ectopic heartbeats. Is the presented method able to discriminate between sinus and arrhythmic beats? I suggest including a quantitative evaluation of this possible ability of the method.
  • Some results are presented in the Method section and this choice makes the section difficult to be read and understood. An example is Figure 2 (very difficult to be read and understand because full of information). I suggest reviewing the section, in particular separating the Method section from the Results section.
  • The results section presents few quantitative measurements of the performance of the method. In order to evaluate the method performance, I suggest a signal-to-noise ratio analysis and a quantification of the sEMG spectral features (as in doi.org/10.1016/j.imu.2018.10.006).
  • The results section presents several comments, that should be placed in the Discussion section.
  • The authors should include the results in the abstract.
  • The authors used both terms “sEMG” and “EMG” for the surface electromyogram. I suggest using only the term “sEMG”.
  • The quality of the figure is low. I suggest reviewing the figure in order to make them more understandable.

Reviewer 2 Report

General comments:

The present manuscript describes several computational algorithms to eliminate cardiogenic artifacts for respiratory sEMG signals, useful in the subsequent estimation of the fatigue level. The authors exhaustively described the applied numerical routines and presented plenty of results supporting their claims (including an Appendix section), which is always welcomed for those readers with more expertise in the field. The main drawback of this study is the use of synthetic signals (simulation study) or derived from CARDIODAT datasets. If authors were able to record sEMG signals from a group of volunteers, why not acquiring at the same time the ECG signal of the volunteers that could produce more realistic ECG influence over the recorded sEMG? At the end, the proposed method(s) by the authors will be judged for patients suffering from more severe cardiogenic comorbidities and in need of respiratory fatigue monitoring. Moreover, the organization of some parts of the manuscript is not satisfactory as it severely disrupts the normal reading process of the manuscript as outlined below.

Technical comments:

. Abstract section: define all the acronyms in this section.

. Line 44: define the acronym for RLS filter.

. Line 50: better say “to each heartbeat aligned at the onset of QRS”.

. Line 74: the authors refer signal upsampling (from 1 kHz to 1024 Hz) but they do not describe the method used to achieve this.

. Figure 1: the units in the y-axis are missing for all the graphs. Also, what are the units for the temporal axis?

. Line 90: first reference to Fig. 1 in the written text. Therefore, reference to subpanel D appears before subpanels A, B and C. Please refer to those before subpanel D in the written text to help improving the readability of the manuscript. Re-order also the figures in the Appendix according to the order they appear first in the written text: A2, A6, etc.

. Line 99: “Raw sEMG signals were high-pass filtered …”

. Equations (3) and (5): Equation 5 is said to be free of ECG contamination, hence the use of the term ECG_SG(t). But for Eq. 3 it contains ECG contamination, right? So, it should be ECG_A(t) or similar instead of ECG_SG(t). Confusing to grasp.

. Line 133: What is the threshold value for QRS identification?

. Lines 141 – 163: I would suggest authors to design a block diagram of the proposed algorithm for raw template construction to help identify the different filtering/processing stages like A3. OK, I found that it corresponds to A4 after reading more through the text. Please inform before.

. Lines 192 – 195: These two last sentences are not understandable in what concerns TSWD35. They seem to contradict each other. Please clarify.

. Line 260: What is GN?

. Line 278: “… as given by Eq. 7”.

. Figure 3: this figure is not readable. Please increase the font size for both axes.

. Line 449: remove 35.

. Figure 4: improve the quality of this figure.

. Line 476: “The goal of this study …”

. Line 550: “the algorithm scored considerably worse than …”

. Appendix figures: missing text references in the main manuscript to Figs. A5 and A9.

Reviewer 3 Report

The manuscript is interesting but its presentation requiresa serious amendments:

  1. Why you use SMR5? Why you use number here?
  2. The word "weaning" is overused in some places.
  3. p. 2, l. 75 1 kHz to 1024 Hz. In my opinion it is wrong. Probably you should have 1 Hz.
  4. p. 5 l. 126 Why you use upsampling? I can't find any explanation to it.
  5. Figure 2, 3, 4 are prepared in a way that you can't see the details. You should remove part of it into supplementary information or make them bigger or reduce to show all details in a clear way.
  6. p. 7 l. 187 why you introduce into the text equation which is too big to fit in the line? The sampe problem is at p.9 l. 301.
  7. There are various methods of ECG analysis including optical method of muscle fatigue detection (muscle tremble detected at reaching lacid acid threshold). Moreover, you can consider similar methods as in a case of EKG signal as well (see e.g.,  SLAMA, Amine B., et al. Application of statistical features and multilayer neural network to automatic diagnosis of arrhythmia by ECG signals. Metrology and Measurement Systems, 2018, 25.1.)

Reviewer 4 Report

The manuscript presents a detailed and comprehensive study of different ECG artifact removal procedures and associated fatigue detection algorithms for EMG signals.

The presented data are massive and a large variety of different algorithms is tested and compared.

The lengthy analysis of results allows to clearly identify strengths and weaknesses of each method combination.

The manuscript, however, has two drawbacks, that should be suitably addressed by the authors.

  1. While the are many clear figures in the supplemental material, the few (only 4!!!) figures present in the main text are really unreadable. The comments in the text are very lengthy and help the reader draw a clear picture of the results, but the picture resulting from figures is at least foggy. The legends and labels are so tiny it is really hard to detect them. The authors are strongly advised to revise the way their results are presented, in order to make figures useful to the reader. They should be probably split in a larger number of figures, with suitable dimensions.
  2. The authors report detailed explanations to justify their choice of resorting to synthetic signals. While I could perfectly agree with most of their reasoning, I still think that at least a few acquisitions of real signals on volunteers would increase confidence on the proposed algorithms. Indeed, it appears that one of the key points is ECG artifact removal and it is not possible to assess how much the synthesized signals can be representative of a real-world scenario. Showing at least the performances as referred to ECG removal on real signals is strongly advisable.

As a last comment, please note that, while the manuscript is generally very well written, a few typos are present here and there.